# Rules and neural nets for morphological tagging of Norwegian
# – Results and challenges

**Dag Trygve Truslew Haug**[1,2], **Ahmet Yıldırım**[1], **Kristin Hagen**[2], **Anders Nøklestad**[2]

[1]Department of Linguistics and Scandinavian Studies, University of Oslo
[2]Humit, University of Oslo

`{daghaug,ahmetyi,kristiha,noklesta}@uio.no`

## Abstract

This paper reports on efforts to improve the Oslo-Bergen Tagger for Norwegian morphological tagging. We train two deep neural network-based taggers using the recently introduced Norwegian pre-trained encoder (a BERT model for Norwegian). The first network is a sequence-to-sequence encoder-decoder and the second is a sequence classifier. We test both these configurations in a hybrid system where they combine with the existing rule-based system, and on their own. The sequence-to-sequence system performs better in the hybrid configuration, but the classifier system performs so well that combining it with the rules is actually slightly detrimental to performance.

## 1 Introduction

The Oslo-Bergen Tagger (OBT, Hagen and Johannessen 2003; Johannessen et al. 2012) is a widely used tool for morphological tagging of Norwegian text. It has existed in various incarnations for around 25 years, first as a purely rule-based system and later coupled with a statistical module for disambiguation. In this paper, we report on our recent efforts to bring the system into the age of neural networks. The question that arises is whether combining the neural system with the existing rules will boost accuracy over a purely neural system. We build two kinds of neural net configurations, one encoder-decoder transformer framework (sequence-to-sequence, seq2seq) and one sequence classification (seqClass) approach. We show that there are challenges in combining rules and neural nets due to divergent tokenisations when the seq2seq approach is employed. In the seqClass approach, on the other hand, the neural net performs so well that combining it with

the rule-based approach is actually detrimental to performance, showing that rule-based methods are not required in the morphological tagging of a language like Norwegian, where a large language model is available (NbAiLab, 2021) and there is sufficient labeled data for fine-tuning. However, we still believe that the rule-based system can be useful for lemmatisation and compound analysis, which we do not consider here.

The structure of the paper is as follows: In section 2 we give some historical background on OBT and in section 3 we describe the current status of its rule-based component. Section 4 describes the training and evaluation data that we have used in developing the new systems. Section 5 then provides the details of how our neural systems were trained. Section 6 describes how they were combined with the rule system. Section 7 evaluates the performance of each of the neural systems on their own as well as in combination with the rules. Section 8 concludes.

## 2 History of the Oslo-Bergen Tagger

The Oslo-Bergen Tagger was originally developed between 1996 and 1998 by the Tagger Project at the University of Oslo. Rules for morphological and syntactic disambiguation were written in the first version of the Constraint Grammar framework (Karlsson et al., 1995), retrospectively called CG1. The input to CG disambiguation rules is multitagged text, i.e., text where each token has been annotated with all possible lexical analyses. Hence, the project also developed a lexicon with lemmas and inflected forms (later known as Norsk ordbank[1]) and a combined tokeniser/multitagger. The tagger was developed for both Bokmål and Nynorsk, the two written varieties of Norwegian. In this article, we will only focus on the Bokmål

---

[1]`https://www.nb.no/sprakbanken/en/resource-catalogue/oai-nb-no-sbr-5/`

version of the tagger, and only on the tokeniser and the morphological disambiguation.

On an unseen evaluation corpus with a wide variety of text genres, OBT achieved an F1-score of 97.2 (Hagen and Johannessen, 2003, 90), with a precision of 95.4 and a recall of 99.0. At the time, this was considered acceptable as the tagger was mostly used to annotate written corpora for linguistic research, where a high recall was considered more important than a high precision.

Since 1998, both the tokeniser and the CG rule interpreter have been changed or modernised several times by different projects (Johannessen et al., 2012). The latest version has an updated lexicon and a tokeniser written in Python which in most cases mirrors the original tokeniser, with the major exception that multiword expressions like *blant annet* ('among other things' - adverb) have been removed from the lexicon and are now dealt with in the CG module. The CG1 rules have been translated to the more efficient and expressive CG3 format and are used with a rule interpreter made by the VISL project at the University of Southern Denmark. Remaining morphological ambiguities and lemma disambiguation are dealt with by a statistical module, implemented as a Hidden Markov Model. This OBT+Stat system achieved an accuracy of around 96% (Johannessen et al., 2012).

## 3 The rule-based tokeniser and tagger

In this section, we first present some of the main tasks for the tokeniser and multitagger before we give a short description of the constraint grammar module. The multitagger uses a lexicon with all possible lexical readings, where a reading is a combination of a lemma and a morphosyntactic tag chosen from a set of 149 possible analyses.[2] The main principle for tokenisation is to split tokens on blank space or a sentence delimiter like a full stop or a question mark. For each token identified, the original word form is looked up in the lexicon. Non-sentence initial capitalised words are identified as proper nouns, while other words that exist in the lexicon are assigned all readings found there. If the word is not found in the lexicon and is not identified as a proper noun, the word is sent to a compound analyser. Most unknown words will get an analysis here, as many of them are productively created compounds. Some words will still

get the tag *ukjent* ('unknown') from the tokeniser. These words are often dialect words not standardised in the lexicon or foreign words. Figure A in the Appendix shows how the tokeniser and multitagger deals with the sentence *TV-programmet "Ut i naturen" begynner kl. 21.15.* ('The TV program "Ut i naturen" starts at 21.15.'), which has quotation marks, abbreviations, and a time expression.

The tokeniser also identifies sentence boundaries using sentence delimiters, a list of known abbreviations and linguistic rules. Headlines are identified by rules as well and get their own tag.

The constraint grammar module takes tokenised and multitagged text as input and its main task is to reduce the number of readings to ideally one per word. The number of readings left by the multitagger varies a lot. In the test corpus used in this article (which will be further described in Section 4) there are on average 2,04 readings per word. After the CG rules are applied, there are on average 1,09 readings left per word.

Figure B in the Appendix shows the output from the CG module in debug mode for the sentence *Rosa cupcakes hører kanskje med når man skal ha bloggtreff?* ('Pink cupcakes might be part of a blog meeting?'). Readings that have been removed start with ";" and the ID numbers of the rules applied are appended to each reading. Note that the English loan word *cupcakes* is not identified in the lexicon or in the compound analyser and has got the tag *ukjent* 'unknown'. The compound *bloggtreff* 'blog meeting' was not in the lexicon but has got two readings from the compound analyser. As the examples show, there are both REMOVE rules (remove a reading) and SELECT rules (select a reading). A rule can be very simple, like rule 2430 in Figure 1 that says "select the verb infinitive reading if the verb to the left is a modal auxiliary and not in the set of dangerous infinitives (= not likely infinitives)".

```
#:2430
SELECT:2430 (verb inf) IF
(NOT 0 farlige-inf)
(-1 m-hj-verb)
;
```

Figure 1: Simple SELECT rule

Figure 2 shows an example of a more complex rule with linked context conditions somewhere to the right in the sentence. The rule says: "choose

---

[2]The complete list is available at `http://tekstlab.uio.no/obt-ny/morfosyn.html`.

the subjunction reading – if somewhere to the right there is a safe noun or pronoun (stop looking if a word on the way has a reading that is not an adverb, adjective or determinative) – and – if there is a word in the present or past tense after the noun/pronoun (adverbs between are fine)."

```
#:2579
SELECT:2579 (sbu) IF
(...)
(**1C subst/pron BARRIER
   ikke-adv-adj-det)
(**1C subst/pron LINK *1
   ikke-adv LINK 0 pres/pret)
;
```

Figure 2: More complex SELECT rule

The CG grammar for Bokmål has more than 2300 rules. 1995 of them are SELECT rules. Some rules apply to all possible words, while some are rules for specific word forms. REMOVE rules look the same as SELECT rules but remove a reading instead of selecting it. During development, we checked the impact of each rule on the recall and precision on a training corpus of 100 000 words from novels, newspapers and magazines before it was added to the rule set.

## 4 Training and evaluation data

The training and evaluation corpora that were used in earlier stages of development of the OBT system are no longer suitable because the tagset and the tokenisation principles have evolved. Instead of bringing this corpus up to date, we chose to use the Norwegian Dependency Treebank (NDT, Solberg et al. 2014) in the development of the latest version of OBT. The Bokmål part of NDT is around 300 000 tokens and consists of blog text, news text, parliament proceedings and government white papers. A problem that we only later became aware of is that most of the raw text contained in the NDT probably went into the Norwegian BERT encoder that we use in our machine learning experiment, which may have caused some data leakage, even if the model did not see the tagged text.

While the principles for annotation in NDT and OBT are close, there are still differences in detail. To ensure that the annotations were as aligned as possible, we ran OBT without statistical disambiguation on the pure text of the NDT and compared the output to the NDT annotations. If the NDT analysis was not among the analyses pro-

duced by OBT, we either corrected the NDT annotation if that was the source of the error, or changed the rules of the OBT system if that could easily be done. This process was iterated a few times. The goal was to improve the quality of the training data for the neural component and to align the output of the OBT with the NDT as the annotation guidelines were slightly different. Also, since the plan was to combine OBT with a neural system, ambiguity reduction by OBT was not a goal in itself if we thought the ambiguity could be resolved by the neural component. Notice that during this period, the whole data set was used for development, which inflates the performance of the rules (and hence the hybrid system we discuss later on) somewhat. But in practice, relatively few changes were made and we did not achieve a full alignment of the annotation guidelines.

The performance of the rule-based system by the end of this phase is shown in Table 1. We see that OBT produces a correct and unambiguous analysis for 90.7% of the tokens and only (one or more) incorrect analyses for 1.8% of the tokens. For 7.5% of the tokens, OBT produces an ambiguous analysis containing the correct tag as one possibility, and the role of the statistical system in a hybrid setup is to pick the correct analysis in these cases. The results are noticeably different from those reported during testing in the nineties (see Section 2), probably because we were not able to fully align the annotation principles of OBT and NDT, and because the precision was calculated differently back then (for example, both the masculine and the feminine reading were regarded as one correct reading for ambiguous feminine and masculine nouns in unmarked contexts).

| result | freq | |
|---|---|---|
| unambiguous correct | 280650 | (90.7%) |
| ambiguous incl. correct | 23219 | (7.5%) |
| wrong | 5413 | (1.8%) |

Table 1: Performance of the rule-based system

Finally, for the neural systems, we split the corpus into train-dev-test sets. While doing this, we applied a simple process for making sure the output tags in the training set covered all output tags in the dev and test sets. The aim is to ensure that the model was trained with samples from all tags. We do this by, first, initializing the Python random seed as 0, then, splitting the data and checking if

the training set covers all tags. If it does not, we increase the random seed by one and do the same until we find a training set that covers all the tags in the other sets. This way, we randomly split the dataset into 80-10-10 percent partitions to obtain train-dev-test datasets respectively.

## 5 The neural systems

Recently, a BERT (Devlin et al., 2018) pre-trained encoder (nb-bert-base) was published by the Norwegian National Digital Library (Kummervold et al., 2021). This pre-trained encoder for Norwegian provides a rich feature set that was previously lacking for the language. Furthermore, since the tagged corpus is very small in comparison to the corpus the pre-trained model was trained on, it is important to use the pre-trained model to be able to generalise to unseen data. We use two different neural system configurations that incorporate this encoder.

### 5.1 The seq2seq configuration

With this configuration, we follow an approach similar to that of Omelianchuk et al. (2020) to tag the sentences using the pre-trained model. Seq2seq models have two main components: an encoder and a decoder. The encoder side is set as the encoder nb-bert-base (NbAiLab, 2021). For the decoder, we randomly initialise 6 layers of size 768 with 12 attention heads. The decoder also has cross-attention layers as it was shown to be effective in seq2seq training (Gheini et al., 2021). We freeze the encoder weights throughout the training since using the encoder as a feature extraction mechanism in this way was shown to be beneficial (Zoph et al., 2016) and is a common practice (Gheini et al., 2021). We use the EncoderDecoderModel provided by the HuggingFace transformers library (Wolf et al., 2020) to configure and train a model.

The encoder-decoder model gets as its input the identifiers of the tokens (token numbers) in the input vocabulary and outputs the token numbers in the output vocabulary. Thus, the input and output are tokenised using these vocabularies. Since the encoder model had already been trained (nb-bert-base) using the widely-utilised sub-word tokeniser Wordpiece (Wu et al., 2016), we use that tokeniser as provided by the Huggingface Tokenizers library. For the decoder side, since our vocabulary size is very small and obvious (82 tags and 5 extra

special tokens such as [CLS] and [SEP]), we do not need to train a special tokeniser. We define the vocabulary manually with these output tokens for use by the Wordpiece tokeniser.

The data were formatted to train the seq2seq network. Figure 3 shows an example of input and output for a sentence. The input is the tokenised form of the sentence. The output is the sequence of serialised tags for each token in the input. The token <next_token> is an indicator that all tags of the corresponding input token have finished and tags of the next input token start afterward.

```
INPUT: Men det er bare noe jeg tror .
OUTPUT:
:konj: clb <next_token>
:pron: 3 ent nøyt pers <next_token>
:verb: pres <next_token>
:adv: <next_token>
:pron: 3 ent nøyt pers <next_token>
:pron: 1 ent hum nom pers <next_token>
:verb: pres <next_token>
$punc$ :clb: <punkt>
```

Figure 3: A sample of sentence input and output for seq2seq training.

The training configuration is as follows: We use the Adam optimiser (Kingma and Ba, 2015) with a learning rate of 0.0001. We set the batch size to 16 sentences as this is the amount the graphic cards could handle. We use the negative log-likelihood loss (Yao et al., 2020) to compute the loss in each batch between the model output and the expected output. For any other parameter not mentioned in this section, we use the default value defined by version 4.17.0 of the Transformers library in the objects of the following types: BertConfig, EncoderDecoderModel, EncoderDecoderConfig, and BertModel.

We evaluate the model using the dev set during the training. We do this by using the BLEU score (Papineni et al., 2002) that is widely utilized to evaluate seq2seq models. We compute the BLEU score between the expected output and the model output for each sentence. We get the average of these scores for the whole dev set. We run the training for 300 epochs and keep the model that results in the maximum average BLEU score for the dev set. To combine the model output with the rules, we use the model's .generate() function (HuggingFace, 2023a) implemented by the library. We set the early_stopping, return_dict_in_generate, and output_scores param-

eters to True. We set num_return_sequences and num_beams to 10 to get the 10 most probable readings given a sentence.

## 5.2 The seqClass configuration

Sequence classification – also referred to as token classification – is a method used to classify a sequence (one or more tokens) into one or more classes such as the type (person, organisation) or sentiment (positive, negative). BERT models have multiple layers that are pre-trained on the language. The Norwegian pre-trained BERT outputs 12 layers (also called hidden states) where each is a 768-dimension vector for each token. Thus, this output is input to a classifier to classify each token. The HuggingFace transformers library (HuggingFace, 2023b) provides a token classification framework that can be used for this purpose. It adds a linear layer on top of the hidden states to make sequence classification possible using the pre-trained encoder.

The dataset has 82 different tags which are used together in different combinations. We observe that the training set has 327 different uses of these combinations.[3] Thus, we treat each combination as a class for this computation. We classify each token into one of these classes which indicates a tagset for that token. We do this by labeling each class as a sequence of zeros and ones where each digit corresponds to one tag. Figure 4 shows an example of tokens and classes of those tokens, where the length of the class names (0's and 1's), which is really 82, is shortened to fit into the figure. The position of 1's in the string indicates the tag assigned to the token. For example, for the first token "Men" the first two columns are assigned which indicate the ":konj:" and "clb" tags (see also Figure 3 for tags of this sentence).

```
Men    1100000000000000000000000000000000
det    0011111000000000000000000000000000
er     0000000110000000000000000000000000
bare   0000000001000000000000000000000000
noe    0011111000000000000000000000000000
jeg    0000100000111000000000000000000000
tror   0000000110000000000000000000000000
.      0000000000000111000000000000000000
```

Figure 4: A sample of a sentence's tokens and their classes used in sequence classification.

[3] In addition to the 149 morphosyntactic analyses (see footnote 2), this includes combinations with various tags that do not convey morphosyntactic information and are ignored during evaluation.

Throughout the training, we use the default parameters defined in the library. We use the Adam optimiser (Kingma and Ba, 2015) with an adaptive learning rate starting from 0.00005. The library uses Cross Entropy Loss (Zhang and Sabuncu, 2018) and picks the model that performs best on the dev set by computing an F1 score. We check the dev set performance for each epoch and run the training for 30 epochs. When combining the model output with the rules, we use the unnormalized final scores of the model (logits) and use torch.topk() to get the topmost 10 probable readings given a sentence.

## 6 Combining neural nets and rules

To combine the output of a neural tagger with the CG tagger, we need to find the intersection of the tag assignments produced by the two taggers. Ideally, we would be able to find such intersections for each individual token separately. However, since the probability of a reading for a particular token depends on the selected readings for all other tokens in the sentence, the only viable option is to consider readings for entire sentences. Thus, for each input sentence, we extract the ten most probable tag assignments produced by the network. Then, for each reading in this list, ordered by decreasing probability, we go through each token and check whether the tag assigned by the network is also found among those left by the CG disambiguation rules. If it is not found, we skip to the next reading in the list. If it is found, we go on to check the next token, and so on until we reach the end of the sentence, at which point the reading is picked as the selected one for the sentence. When the tokenisations are different, it is not clear what to do. But if the tokens are the same, but the intersection of the sets of possible tags left by the CG system and the neural net is empty, we can default to the most probable reading in the neural net output.

Figure 5 shows a case where the tokenisation of the seq2seq neural system does not match with that of OBT. The neural system has split the initial, unknown proper name at a hyphen, whereas the CG tagger keeps it as one token. Since tokenisation is part of a preprocessing step and misalignments in tokenisation is a problem to be solved separately from tag assignment, we simply disregard such sentences in the evaluation. However, it should be noted that this problem is only acute for

the seq2seq system, which produced mismatching tokenisation in 205 out of 2003 sentences (10.2%). For the seqClass system, the problem is smaller: 57 out of 2003 sentences (2.8%).

```
Neural net: Garosu - gil , som betyr [...]
CG: Garosu-gil , som betyr [...]
```

Figure 5: Mismatching tokenisation

Figure 6 shows the problem of mismatching tags. For the first word, the CG tagger has left five possible analyses, and the neural net has correctly disambiguated to the plural adjective reading. However, OBT did not recognise the second word, *cupcakes*, and has therefore left an *ukjent* ('unknown') tag while the neural system has no analysis with that tag. Notice that the figure only shows the neural system's most probable assignment of tags to the whole sentence. The actual output is a probability distribution over tag assignments, but in this case, no probability was assigned to any tag assignment containing the *ukjent* tag for *cupcakes*, which is the only analysis produced by the rule system.

```
Neural net:
Rosa        adj fl pos
cupcakes    subst appell fl mask ub  <---
hører       verb pres
kanskje     adv
med         prep
når         sbu
man         pron ent hum nom pers
skal        verb pres
ha          verb inf
bloggtreff  subst appell ent nøyt ub
?           clb <spm>

CG:
Rosa        adj fl pos
            adj nøyt ub ent pos
            adj ub m/f ent pos
            subst appell ubøy
            subst appell fem be ent
cupcakes    ukjent                   <---
hører       verb pres
kanskje     adv
med         prep
når         sbu
man         pron ent hum nom pers
skal        verb pres
ha          verb inf
bloggtreff  subst appell ent nøyt ub
            subst appell fl nøyt ub
?           clb <spm>
```

Figure 6: Non-intersecting tags

For such cases, we default to the most probable analysis generated by the neural net. This is not necessarily the best option: as we will see in

| system | accuracy |
|--------|----------|
| pure seq2seq | 92.71 |
| seq2seq + OBT | 94.15 |
| pure seqClass | 100.0 |
| seqClass + OBT | 99.99 |

Table 2: Accuracy of different systems

Section 7, the seq2seq system is often incorrect in cases where the tag assignments do not intersect. Moreover, the problem of mismatching tag assignments is quite common, happening in 386 out of the 2003 sentences (19.3%).

In the seqClass system, non-intersecting tag assignments are even more frequent, at 466 sentences (23.3%). However, as we will see in the next section, the neural net in this configuration is more precise than the rules, so that defaulting to its output yields the correct reading.

# 7 Evaluation and error analysis

We evaluate both the seq2seq system and the seqClass system on their own and as combined with the rule-based system in the way described in Section 6. This yields four different systems. The performance of the four systems is shown in Table 2.

These numbers are only computed over sentences where the tokenisation matches. This means that the seq2seq system, in both its pure and hybrid form, is tested only on sentences where the seq2seq system, the OBT tagger and the gold agree on the tokenisation. As we saw in section 6, this means that 10.2% of the test data are left out. It would have been possible to test the pure seq2seq system on the sentences where its tokenisation agrees with the gold, without considering what OBT does, but since we want to compare the pure neural system to the hybrid system, we held the evaluation set constant between these two setups. Similarly, for the seqClass system, we left out the 2.8% of sentences where either OBT or the neural system had tokenisation that does not match the gold for both the pure and the hybrid system. Notice that this means the seqClass system is tested on a larger set of sentences than the seq2seq system.

Overall, we see that the seqClass system performs best and in fact achieves a perfect score. This is of course a rather debatable result, which we will look into in section 7.2. But notice that the 2.8% of sentences with diverging tokenisations

are incomparable and therefore not evaluated here, though they could obviously be considered errors of the system.

## 7.1 The seq2seq system

The seq2seq system performs reasonably well on its own, but clearly benefits from being intersected with the rules, yielding a 1.3% accuracy boost to 94.1%. By contrast, the widely used Spacy tagger reports an accuracy of 95.0% for morphological tagging of Norwegian UD.[4]

Most of the errors in this setup comes from the fact that we default to the best neural analysis when there is no intersection. As it turns out, the neural system is wrong in most of these cases. If we restrict attention to only sentences where the tags intersect (70.5% of the total), accuracy is at 99.0%. Put another way, when we reduce the test set in this way, its size decreases by 8036 tokens from 26648 to 18612, but the number of errors decreases from 1940 to 565. This indicates an error rate of 17.1% on the tokens in sentences where the intersection of the tag assignments from the neural system and the CG tagger is empty.

Turning now to the kind of errors the seq2seq system makes, we show the twelve most common error types of the pure and the hybrid system in Table 3 and Table 4 respectively. We see that the most common error is mixing up the distinction between neuter and common gender adjectives, which in many cases is not expressed morphologically. Other than that, most errors involve either over- or underpredicting the tag :prep: (preposition). This error source is somewhat reined when the system is interfaced with the rules. But many errors of this kind remain, either because this analysis is also suggested by the rules and so picked as the most probable tag, or more likely because there is no intersection between the tag assigments, i.e. neither :prep: nor any other tag suggested by the neural system is among the tags left by the rule-based system.

Overall, this confusion around the :prep: tag seems a distinct deficit of the seq2seq model. Other errors, such as those involving gender, or the number of indefinite neuter nouns (which make no morphological singular/plural distinction), or the identification of perfect participles which of-

---

[4]See https://spacy.io/models/nb. As the Norwegian UD corpus (Øvrelid and Hohle, 2016) is an automatic conversion of the NDT corpus, the complexity of the tasks should be comparable, although the test split is not identical.

ten co-exist with homonymous adjectives in Norwegian (as in other Germanic languages, cf. English 'bored') are more as one would expect from any system because there might not be enough signal in the training data to pick up the distinctions, which often depend on subtle properties of the context. However, what we observe here is that intersecting with the rules actually worsens the accuracy. The hybrid system overapplies the adjective analysis in two different varieties for a total of 17+13 errors. By contrast, in the pure seq2seq setup, this error is not frequent enough to figure in the table. It does occur in 21 cases, but that is still notably less than in the hybrid system. This shows that the CG rules wrongly disambiguate these cases, which is not surprising since the distinction as made according to the NDT guidelines relies on semantic distinctions. It would be hard to tune the CG rules to those distinctions, and we did not make any attempt at that, but there seems to be enough signal in the data for the seq2seq system to pick it up to some extent.

## 7.2 The seqClass system

The seqClass system achieves a suspicious, perfect score on our test set when used alone, and makes one error when combined with the rules. This error is instructive in itself: it concerns a single-word "sentence", namely the heading "Justisdepartementet" ('The Department of Justice'). The CG tagger considers this a common noun. This is only the fifth most probable tag according to the neural net, but it is among the possibilities and so it is chosen by the hybrid system, although the gold considers it a proper noun.

This is the only instance of such an error. In other words, the seqClass system not only assigns the highest probability to the correct tag in every case, but also performs well enough as to not rank any incorrect suggestions by the CG system among the top 10 readings that we consider for intersection, except in this one case.

The looming question is of course how the system manages to perform so well. Some degree of overfitting must have taken place, but can hardly explain everything. Moreover, as we noted in Section 4, it is likely that all or at least most of the raw text of NDT went into the Norwegian BERT model, which may have caused some data leakage. More worryingly, we cannot completely exclude the possibility that the language model has been

| Gold tag | Predicted tag | Freq |
|---|---|---|
| [':adj:', 'ent', 'nøyt', 'pos', 'ub'] | [':adj:', 'ent', 'm/f', 'pos', 'ub'] | 24 |
| [':subst:', 'appell', 'ent', 'mask', 'ub'] | [':prep:'] | 24 |
| [':prep:'] | [':subst:', 'appell', 'ent', 'mask', 'ub'] | 19 |
| [':verb:', 'pres'] | [':prep:'] | 18 |
| [':subst:', 'appell', 'ent', 'mask', 'ub'] | [':subst:', 'appell', 'ent', 'fem', 'ub'] | 18 |
| [':prep:'] | [':subst:', 'prop'] | 17 |
| [':subst:', 'appell', 'ent', 'fem', 'ub'] | [':subst:', 'appell', 'ent', 'mask', 'ub'] | 16 |
| [':prep:'] | ['$punc$', ':<komma >:'] | 15 |
| [':prep:'] | [':verb:', 'pres'] | 14 |
| [':subst:', 'appell', 'fl', 'mask', 'ub'] | [':subst:', 'appell', 'fem', 'fl', 'ub'] | 14 |
| [':subst:', 'mask', 'prop'] | [':subst:', 'prop'] | 14 |
| [':subst:', 'appell', 'ent', 'mask', 'ub'] | [':subst:', 'appell', 'ent', 'nøyt', 'ub'] | 14 |

Table 3: Most frequent errors, pure seq2seq system

| Gold tag | Predicted tag | Freq |
|---|---|---|
| [':adj:', 'ent', 'nøyt', 'pos', 'ub'] | [':adj:', 'ent', 'm/f', 'pos', 'ub'] | 22 |
| [':subst:', 'appell', 'ent', 'mask', 'ub'] | [':subst:', 'appell', 'ent', 'fem', 'ub'] | 18 |
| [':subst:', 'appell', 'ent', 'mask', 'ub'] | [':prep:'] | 18 |
| [':prep:'] | [':subst:', 'appell', 'ent', 'mask', 'ub'] | 17 |
| [':verb:', 'pres'] | [':prep:'] | 17 |
| [':verb:', 'perf-part'] | [':adj:', '<perf-part>', 'ent', 'm/f', 'ub'] | 17 |
| [':prep:'] | [':subst:', 'prop'] | 16 |
| [':verb:', 'perf-part'] | [':adj:', '<perf-part>', 'ent', 'nøyt', 'ub'] | 13 |
| [':subst:', 'appell', 'fl', 'mask', 'ub'] | [':subst:', 'appell', 'fem', 'fl', 'ub'] | 13 |
| [':prep:'] | [':verb:', 'pres'] | 12 |
| [':subst:', 'appell', 'ent', 'nøyt', 'ub'] | [':subst:', 'appell', 'fl', 'nøyt', 'ub'] | 12 |
| [':verb:', 'perf-part'] | [':prep:'] | 11 |

Table 4: Most frequent errors, hybrid seq2seq system

exposed to the CONLL file (and hence the manually corrected tags), although it seems unlikely. In any case, we would have expected some errors in the tokens where we changed the analysis. Moreover, none of these factors would explain why the model is also able to assign a very low probability to the incorrect suggestions from the CG.

We plan to conduct a more thorough test of the system on recent text which the BERT model cannot have been exposed to. So far we have only been able to conduct a very preliminary test. We downloaded web text from nrk.no (the Norwegian national broadcaster) from 2023, i.e. after the Norwegian BERT was published. This text was tagged both with the original system of OBT + HMM-based disambiguation, and with the new seqClass system. For the first 2000 tokens, we inspected all mismatches between the two systems, on the (questionable) assumption that whenever the two systems agree, the tag is likely correct. We found 144 discrepancies, and by manual judgement 137 were considered errors by the old system, and 7 were considered errors by the pure seqClass system. This evaluation method is obviously not perfect, but it does suggest that the pure seqClass sys-

tem makes very few errors. Further proper evaluation must follow, but the results are clear enough to discourage future work on the rule-based system.[5]

# 8 Conclusion

We have presented our efforts to improve the Oslo-Bergen tagger for Norwegian morphological tagging. Two neural systems were trained, based on a sequence-to-sequence setup and a sequence classifier setup, both built on top of the Norwegian BERT model of Kummervold et al. (2021). Both were tested on their own and in combination with the rule-based OBT system. The sequence-to-sequence system did not outperform earlier benchmarks on its own, but improved when combined the rules. However, the sequence classification setup was much better and in fact achieved a surprising perfect score on the test set. While we will explore the causes of this, preliminary testing on new data supports the conclusion that the new system makes very few errors, and we will focus on validating this in a more proper evaluation setting.

[5]The seqClass model is available for download at https://github.com/textlab/norwegian_ml_tagger.

## Acknowledgments

This work was supported by the CLARINO project, funded by RCN FORINFRA grant no. 295700, and the Universal Natural Language Understanding project, funded by RCN IKTPLUSS grant no. 300495.

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

## Appendix: sample multitagger and CG output

```
<word>Tv-programmet</word>
"<tv-programmet>"
"tv-program" subst appell nøyt be ent
 samset-leks <*program> <+programmet>
<word>«</word>
"<«>"
"$«" <anf>
<word>Ut</word>
"<ut>"
"ut" prep
"ut" adv
<word>i</word>
""
"i" prep
"i" subst appell mask ub ent
<word>naturen</word>
"<naturen>"
"natur" subst appell mask be ent
<word>»</word>
"<»>"
"$»" <anf>
<word>begynner</word>
"<begynner>"
"begynne" verb pres
"begynner" subst appell mask ub ent
<word>kl.</word>
"<kl.>"
"kl." subst appell fork
<word>21.15</word>
"<21.15>"
"21.15" subst <klokke>
"21.15" det kvant
<word>.</word>
"<.>"
"$." clb <<< <punkt> <<<
```

Figure A: Tokenised and multitagged sentence

```
<word>Rosa</word>
"<rosa>"
"rosa" adj fl pos
"rosa" adj nøyt ub ent pos
"rosa" adj ub m/f ent pos
"rosa" subst appell ubøy
"rose" subst appell fem be ent
; "rosa" adj be ent pos REMOVE:2311
<word>cupcakes</word>
"<cupcakes>"
"cupcakes" ukjent
<word>hører</word>
"<hører>"
"høre" verb pres
<word>kanskje</word>
"<kanskje>"
"kanskje" adv
<word>med</word>
"<med>"
"med" prep
<word>når</word>
"<når>"
"når" sbu SELECT:2579
; "nå" verb pres SELECT:2579
; "når" adv REMOVE:3383
<word>man</word>
"<man>"
"man" pron ent pers hum nom
 SELECT:3451
; "man" subst appell fem ub ent
 SELECT:3451
; "man" subst appell mask ub ent
 SELECT:3451
; "mane" verb imp SELECT:3451
<word>skal</word>
"<skal>"
"skulle" verb pres <aux1/perf_part>
 <aux1/infinitiv>
<word>ha</word>
"<ha>"
"ha" verb inf <aux1/perf_part>
 SELECT:2430
; "ha" interj SELECT:2430
; "ha" subst symb REMOVE:3574
; "ha" verb imp <aux1/perf_part>
SELECT:2430
<word>bloggtreff</word>
"<bloggtreff>"
"bloggtreff" subst appell nøyt ub ent
 samset-analyse <+treff>
"bloggtreff" subst appell nøyt ub fl
 samset-analyse <+treff>
<word>?</word>
"<?>"
"$?" clb <<< <spm> <<<
```

Figure B: Tokenised, multitagged and disambiguated sentence