# OpenReview forum: "Integrating rules and neural nets for morphological tagging of Norwegian - Results and challenges "
_NoDaLiDa/2023/Conference — NoDaLiDa 2023_

### Official Review · Reviewer_vxPL · 2023-02-19

**Rating:** 3
**Confidence:** 4

**Review:**

This paper describes an effort in combining a rule-based tagger with a neural network-based tagger for Norwegian morphological tagging. Empirical results are given comparing the neural and combo taggers. The authors note issues in tokenization as a complicating factor in the combination.

I think this idea is good, and reusing the strengths of old rule-based systems is a good idea. But I am unfortunately not able to understand some key points.

Unfortunately, I am not able to understand how the combination is actually done. In my understanding, the rule-based system (OBT) outputs a set of tags for each word, whereas the neural system (NN) outputs a single tag for each word. The combination works by taking the intersection of words in those cases where there is a non-empty intersection, and the NN suggestion otherwise. I just cannot understand how this does not just equal just taking the NN prediction. According to the results, using the rules somehow improves over using just the NN tagger. I am unfortunately unable to see how this works. If the NN tagger outputs more than one tag, it should be described how that happens, and if so, how you select over intersections with more than one tag. This is a key point of the paper.

I find it a bit strange that you use a seq2seq model for the neural network, and evaluate using BLEU during training (rather than accuracy as you. do in your final evaluation). Why do you not use a more standard classification or multi-classification setup? I think then you could more easily have solved the tokenization issue as well, by using standard schemes for subword tokens, such as averaging or using the first subword.

I would like to see some more detailed statistics of your changes to NDT. How often did you change the OBT, and how often did you change the gold standard? Do you plan to release the new NDT gold standard?






**Paper Type:**

Long paper

---

### Official Review · Reviewer_HGEb · 2023-03-09
**Integrating rules and neural nets for morphological tagging of Norwegian - Results and challenges**

**Rating:** 7
**Confidence:** 3

**Review:**

The paper reports on approaches for and results from modernizing the Oslo-Bergen Tagger (OBT) for Norwegian morphological tagging, primarily by upgrading the statistical module from being HMM-based to a DNN-based module.
OBT has been around for a quarter century, and it is nice that the paper includes an overview of the development and variants of the system over time.

The motivation for replacing the older HMM module is twofold. It is remarked that a weakness of the HMM approach was that

>the HMM might select a tag that was previously removed by the disambiguation rules or not even present in the tagger lexicon

This indicates that the HMM output only the most likely hypothesis. It should be noted that several approaches exist to enable HMM-based systems to output multiple hypotheses, either as lists or as lattices, including relevant probability scores. In other words, it could have been possible to alleviate this weakness also with HMM technollgy. However, the second part of the motivation, that DNNs have demonstrated superior performance on a number of NLP tasks is undeniable.

The novelty of the paper is in the implementation of the neural model and the integration of this with the rule-based part of the system, which is interesting but also demonstrates some of the difficulties in combining two rather different approaches/philosophies. The approach is not particularly original, but is reasonable and sensible.

The paper is generally clearly written.

Some more detailed comments:
The text includes examples of SELECT rules. It would have been good to also include an example of the REMOVE rules

In the description of the development of the new version of the rule-based part of OBT, it is stated that
>Notice that during this period, the whole data set was used for development, as is common with rule-based systems

 If this practice is indeed common (I do not personally work with rule-based systems), I am appalled. If the evaluation data is part of the training data, how can an assessment of the system be valid for any unseen data? Moreover, in the later stages of the current paper, the corpus is divided into train/dev/test partitions. If the rule-based component has been exposed to the test partition, the evaluation of the hybrid system is severely flawed.



The procedure for splitting of the corpus into train/dev/test parts (lines 343-357) is constructed

>the output tags in the training set covered all output tags in the dev ant test sets to ensure that the model was trained with samples from all tags

This appears to ensure at least a single sample of all tags would be in the training set, however, a single sample is not sufficient for training a statistical model.


**Paper Type:**

Long paper

---

### Official Review · Reviewer_RtLY · 2023-03-09
**The paper presents a clear report on how one could combine older and newer software for achieving better results in morhological tagging.**

**Rating:** 7
**Confidence:** 4

**Review:**

Strength of the paper is its clarity.

The description of the corpora, tagging methods, tagset and errors are detailed enough to be understandable.
For someone who would also like to combine a rule-based POS tagger with a neural model (for another language perhaps), the paper might serve as a model to follow. The historical overview is also really useful, as it explains the motivation for re-doing work that has been done and re-done earlier already.

The weakness of the paper is its lack of novelty.
Combining taggers is a well known task that has been explored before for various languages and methods.

The target audience of the paper are engineers and practitioners who need to provide robust high-quality tools that get the job done.

Summary:

The paper presents a clear report on how one could combine older and newer software for achieving better results in morhological tagging.


**Paper Type:**

Long paper

---

### Decision · Program_Chairs · 2023-03-17

Accept